# Computational Screening of Newly Designed Compounds against Coxsackievirus A16 and Enterovirus A71

**DOI:** 10.3390/molecules27061908

**Published:** 2022-03-15

**Authors:** Amita Sripattaraphan, Kamonpan Sanachai, Warinthorn Chavasiri, Siwaporn Boonyasuppayakorn, Phornphimon Maitarad, Thanyada Rungrotmongkol

**Affiliations:** 1Structural and Computational Biology Research Unit, Department of Biochemistry, Faculty of Science, Chulalongkorn University, Bangkok 10330, Thailand; bbeeamita@gmail.com (A.S.); sanachaikamonpan@gmail.com (K.S.); 2Department of Chemistry, Faculty of Science, Chulalongkorn University, Bangkok 10330, Thailand; warinthorn.c@chula.ac.th; 3Applied Medical Virology Research Unit, Department of Microbiology, Faculty of Medicine, Chulalongkorn University, Bangkok 10330, Thailand; siwaporn.b@chula.ac.th; 4Research Center of Nano Science and Technology, Shanghai University, Shanghai 200444, China; pmaitarad@shu.edu.cn; 5Ph.D. Program in Bioinformatics and Computational Biology, Graduate School, Chulalongkorn University, Bangkok 10330, Thailand

**Keywords:** hand foot and mouth disease, coxsackievirus A16, enterovirus A71, 3C protease, in silico drug design

## Abstract

Outbreaks of hand, foot, and mouth disease (HFMD) that occur worldwide are mainly caused by the Coxsackievirus-A16 (CV-A16) and Enterovirus-A71 (EV-A71). Unfortunately, neither an anti-HFMD drug nor a vaccine is currently available. Rupintrivir in phase II clinical trial candidate for rhinovirus showed highly potent antiviral activities against enteroviruses as an inhibitor for 3C protease (3Cpro). In the present study, we focused on designing 50 novel rupintrivir analogs against CV-A16 and EV-A71 3Cpro using computational tools. From their predicted binding affinities, the five compounds with functional group modifications at P1′, P2, P3, and P4 sites, namely P1′-1, P2-m3, P3-4, P4-5, and P4-19, could bind with both CV-A16 and EV-A71 3Cpro better than rupintrivir. Subsequently, these five analogs were studied by 500 ns molecular dynamics simulations. Among them, P2-m3, the derivative with meta-aminomethyl-benzyl group at the P2 site, showed the greatest potential to interact with the 3Cpro target by delivering the highest number of intermolecular hydrogen bonds and contact atoms. It formed the hydrogen bonds with L127 and K130 residues at the P2 site stronger than rupintrivir, supported by significantly lower MM/PB(GB)SA binding free energies. Elucidation of designed rupintrivir analogs in our study provides the basis for developing compounds that can be candidate compounds for further HFMD treatment.

## 1. Introduction

Hand, foot, and mouth disease (HFMD) is one of the global public health concerns that are widely spread worldwide, especially in the southeast pacific region, e.g., China [1], Japan [2], Taiwan [3], Singapore [4], and Thailand [5]. It is easily transmitted by direct contact through excretion, saliva, and fecal matter [6]; thus, the number of people infected with HFMD has increased almost every year, particularly children under five years old [7,8,9]. HFMD is generally associated with the viral infection of coxsackievirus A16 (CV-A16) and enterovirus A71 (EV-A71), which belong to the Enterovirus genus of Picornavirales order [10]. Moreover, the EV-A71 strongly correlated with the more severe clinical outcome, especially in neurological sequelae such as encephalitis and meningitis [11,12]. The genome of enteroviruses contains a positive single-strand RNA (ssRNA) with a single open reading frame (ORF) encoding a large polyprotein precursor that requires proteolytic processing to produce viral structural and replication proteins. After the virus enters the host cells, a viral polyprotein is produced and further cleaved into the four structural proteins (Vp1, Vp2, Vp3, and Vp4) and seven non-structural proteins (2A, 2B, 2C, 3A, 3B, 3C, and 3D) by the activity of viral and host proteases [13]. The 3C cysteine protease (3Cpro, Figure 1A) favorably cleaves the scissile peptide bond between glutamine (Q) and glycine (G) through its catalytic residues (H40, E71, and C147) during the viral replication process. In addition, the EV-A71 3Cpro facilitates progeny virus production and helps the virus evade host antiviral immunity by interaction with the cleavage of host factors [14]. Therefore, the primary roles of 3Cpro in the life cycle of EV-A71 and CV-A16 make it an ideal drug target against CV-A16 and EV-A71 viruses [13,14].

Rupintrivir (AG7088, chemical structure in Figure 1B) is a drug candidate against the 3Cpro of human rhinovirus (HRV) [13] and is currently proven to be the most effective peptidomimetic 3Cpro inhibitor with a half-maximal inhibitory concentration (IC_50_) of 2.1 nM against CV-A16 [14,15]. In addition, it displays a broad-spectrum inhibitory activity against other viruses belonging to the *Piconarviridae* family, such as CVB2, CVB5, EV6 EV-A71, and EV9 [16,17]. Its activity significantly decreases in EV-A71 approximately ~100-fold compared to HRV [7,10,18]. The rhinovirus inhibition in the phase II clinical trial was stopped because the rupintrivir failed to meet desired clinical parameters [19,20,21,22,23,24]. Moreover, it was poorly aqueous soluble with low oral bioavailability challenging further pharmacological development [25]. In addition, the synthesized peptidic Michael acceptor compound SG85 is a rupintrivir-modified compound that acts as an inhibitor of EV-A71 3Cpro (EC_50_ of ~180 nM) [26]. A previous study of molecular dynamics (MD) simulations revealed that SG85 shared a binding pattern against CV-A16 3Cpro similar to the rupintrivir/EV-A71 complex [27]. According to the study of Wang et al. (2017), N69 residue in the active site has been reported to stabilize the S2 pocket of EV-A71 3Cpro by forming a hydrogen bond with N atoms of L70 and E70 residues. The mutation of N69 abolishes the bond network by destabilizing the S2 pocket. Thus, a natural substrate binding to EV-A71 3Cpro can possibly occur in the presence of an inhibitor. They suggested that it is conceivable that modification of the P2 residue with a longer side chain can increase the inhibitory effect [28].

For the drug design and development process, the cost and time consumption in high throughput screening for hit (lead) compounds is generally expensive; however, it can be reduced by applying high-performance computational techniques. In this work, the interaction energy prediction and all-atom molecular dynamics (MD) simulation [29] were employed to screen a series of the designed rupintrivir analogs against 3Cpro of CV-A16 and EV-A71. Additionally, detailed knowledge of the binding mechanisms of the most potent compound would be helpful in the development of new anti-HFMD agents.

## 2. Results and Discussion

### 2.1. Rational Design and Screening

Based on the inhibitor–ligand interactions of rupintrivir binding to EV-A71 3Cpro in Appendix A, the 50 analogs were modified using a structure-based drug design as follows. P1 was modified at O^3^ (e.g., chlorine, fluorine, and methanol) to better interact with K143 in the S1 subsite. P2 was enlarged by the bulkier side chain (e.g., methanamine, ethylamine, and ethanol) to consequently shorten the distance to K130 and Q71 residues in the S2 subsite. To interact with S128 in the S3 subsite, the addition of 2-propanol, 2-fluoropropane, or ammonia was introduced on the side chain of P3. As L125 is located in the S4 subsite, we decided to increase the length of the P4 side chain or to change the functional group (e.g., hydroxyl, methyl, and fluorine). The chemical structures of all 50 compounds are given in Appendix A.

The previous study reported that EV-A71 was the cause of severe and fatal cases of HFMD (90%), while non-EV-A7 enteroviruses were associated with less than 10% of severe and fatal cases [30]. Therefore, the EV-A71 was used as a reference protein for the initial energy filtering. The MM/PB(GB)SA interaction energy calculations were applied on the minimized complex of analogs/EV-A71 3Cpro. The relative interaction energy of each complex compared to the parent compound rupintrivir ($\Delta \Delta G_{bind}=\Delta G_{bind}^{analog}-\Delta G_{bind}^{rupintrivir}$) is shown in Table 1. Among the 50 designed rupintrivir analogs, the five compounds P1′-1, P2-m3, P3-4, P4-5, and P4-19 with negative $\Delta \Delta G_{bind}$ (2D structure shown in Figure 2) were selected for investigating the binding pattern and interaction profile in EV-A71 and CV-A16 3Cpro by MD simulations in a further step.

### 2.2. Stability of the Simulated Complexes

The root mean square displacement (RMSD) of all atoms for each system relative to the minimized structure versus simulation time was measured and plotted in Figure 3. The RMSD values of the complexation between rupintrivir or its five analogs and EV-A71/CV-A16 3Cpro from the three independent simulations were about 1.0–2.0 Å from the beginning of simulation until the end. In addition, the superimposition of compounds P1′-1, P2-m3, P3-4, P4-5, P4-19, and rupintrivir against CV-A16 and EV-A71 at the binding site derived from the last 50 ns of simulation in run1 were performed (Appendix A). This finding suggested that all ligands were likely stable along with the simulations in the active site. In the study, the last 50 ns of all three simulations were considered for further analyses regarding the number of contact atoms and intermolecular hydrogen bonds between compound and protein targets.

### 2.3. Number of Contact Atoms and H-Bonds

To evaluate the binding strength of the designed compounds in the active site of 3C pro, the number of atom contacts within the 3.5-Å sphere of each analog were counted. The average numbers of contact atoms in the last 50 ns from the three-independent simulation are summarized in Table 2. Among five designed analogs, the P2-m3 showed the highest number of contact atoms in both CV-A16 (24.3 ± 4.3) and EV-A71 (22.1 ± 4.5) systems. In addition, this compound gave the number of contact atoms higher than rupintrivir (23.7 ± 4.7 for CV-A16 and 20.0 ± 5.0 systems).

The hydrogen bond (H-bond) formation is one of the essential factors that can determine the binding strength of the interactions between inhibitors and surrounding amino acid residues at the enzyme active site. The intermolecular hydrogen bonds were calculated using the two criteria, i.e., the distance between hydrogen donor (HD) and hydrogen acceptor (HA) ≤ 3.5 Å, and the angle of HD-H⋯HA ≥ 120°. The average numbers of H-bond at the last 50 ns from three independent simulations are given in Table 3. Again, we found that P2-m3 showed the highest number of hydrogen bonds in CV-A16 and EV-A71 systems (6.5 ± 1.2 and 5.6 ± 1.5), which were more than rupintrivir (4.5 ± 0.9 and 4.7 ± 1.1). These findings suggested that P2-m3 fitted well within the binding pocket of EV-A71 and CV-A16 3Cpro.

By considering the binding pattern in terms of hydrogen bond of P2-m3 in comparison with rupintrivir, the intermolecular H-bonds formed with 3Cpro of EV-A71 and CV-A16 are plotted and illustrated in Figure 4 and Figure 5. It can be seen that rupintrivir was stabilized within CV-A16 and EV-A71 3Cpro by forming four H-bonds with the residues in two pocket sites: (i) at P1 site, H161, I162, and K143 residues with O^3^, N^2,^ and N^1^ atoms (see atomic labels in Figure 1B) and (ii) at P3 site, G164 residue with O^5^ atom. This finding corresponded to the rupintrivir/CV-A16 complex from the X-ray structure [13] and the previous MD study [27]. For the analog P2-m3, the introduction of aminomethyl substitution in meta-position at the P2 site raised the H-bond formation with L127 and K130 in EV-A71 (47.7% and 61.5%) and CV-A16 (40.7% and 77.7%). A medium H-bond with G164 was also detected in CV-A16 3Cpro at the P3 site.

### 2.4. Key Binding Residues

The calculation of per-residue free energy decomposition based on the MM-PBSA method was applied on the 50 frames from the last 50 ns of the three simulations (150 structures in total) to study the critical residues for ligand binding to 3Cpro of CV-A16 and EV-A71. The results are given in Figure 6 and Figure 7, where only residues that exhibit the energy stabilization of ≤−0.5 kcal/mol are labeled and discussed. The key residues binding of CV-A16 3Cpro with rupintrivir were H40, L125, L127, T142, A144, C147, I162, G163, G164, N165, and F170 residues. Likewise, the additional residues K143, G145, Q146, and H161 were in the EV-A71 system. The obtained results were consistent with the previous work [13]. Although the binding pattern of P2-m3 in both targets was likely similar to its template rupintrivir, the residue contribution for P2-m3 binding was more pronounced. Additionally, it was also stabilized by the additional residues F25, N126, S128, and K130 in EV-A71 and F25, S128, K130, K143, Q146, and H161 in CV-A16.

### 2.5. Predicted Binding Affinity of the Potent Rupintrivir Analog

The binding efficiency of the newly designed compound P2-m3 with EV-A71 and CV-A16 3Cpro was estimated by the MM/(GB)PBSA method on the same set of snapshots used in the per-residue free energy decomposition calculation (Appendix A). The molecular mechanics energy components in the gas phase ($\Delta E_{\mathrm{MM}}$) and binding free energy based on the MM/PBSA method $(\Delta G_{\mathrm{bind}}$) results of each system are depicted in Figure 8. The result showed that the P2-m3 had a stronger binding affinity than rupintrivir by ~8 and ~3 kcal/mol in EV-A71 and CV-A16 (Figure 8B) due to a stronger electrostatic attraction (Figure 8A). Our finding agree with the previous study, which suggested that modification of the P2 residue with a longer side chain can increase the possibility that inhibitor will bind to EV-A71 3Cpro leading to increased inhibitory effect [28]. Changing the fluorobenzyl group at the P2 site of rupintrivir to the aminomethyl-benzyl group in P2-m3 could enhance the ligand-binding affinity in both proteases. Moreover, P2-m3 showed better solubility than rupintrivir from the result of ADMET property (Appendix A).

## 3. Materials and Methods

### 3.1. System Preparation and Compound Screening

The X-ray crystal structures of rupintrivir in complex with the 3Cpro of CV-A16 and EV-A71 were obtained from Protein Data Bank (PDB), entry codes 3SJI [13] and 3R0F [14], respectively. Note that the CV-A16 and EV-A71 3Cpro share a similar sequence with 91% of identity and 96% of similarity (Appendix A). Based on inhibitor–target interactions in CV-A16/EV-A71 3Cpro complex, the structure-based drug design was used to design the rupintrivir analogs at P1′, P1, P2, P3, and P4 sites. To prepare the 3D structures of each designed ligand, their protonation states were then determined using PROPKA 3.1 [31]. The partial atomic charges and empirical force field parameters for each ligand were developed according to the standard procedure [32,33,34]. The atomic charges of each inhibitor were calculated using HF/6–31G(d) method implemented in the Gaussian09 software [35]. The electrostatic potential (ESP) charges were consequently calculated with the same level of theory and were then fitted into restrained ESP (RESP) charges using the ANTECHAMBER module of AMBER16 [36,37]. The FF14SB [38] and GAFF2 [39] force fields were applied for protein and ligands, respectively. All missing hydrogen atoms of protein and ligand were added using the LEaP module and were then minimized to remove the bad contacts. The complexes were solvated in the TIP3P [40,41] water box with a minimum distance of 10 Å between the protein surface. Afterward, the complexes were energy-minimized by 1500 interactions of steepest descent (SD) and conjugated gradient (CG) methods using AMBER16 with the AMBER ff14SB force field. The binding affinity of all designed analogs toward both 3Cpro enzymes was predicted using MMPB(GB)SA interaction energy calculations. The designed ligands with lower interaction energy than rupintrivir were selected to study the structural dynamics and binding strength within proteins by all-atom molecular dynamics simulations.

### 3.2. Molecular Dynamics Simulations

The potent ligands from interaction energy screening were simulated under periodic boundary conditions with NPT ensemble. In brief, a residue-based cutoff of 10 Å was employed for nonbonded interactions, and the particle mesh Ewald summation method [42] was used to treat the electrostatic interactions. The SHAKE algorithm [26] was applied to constrain all covalent bonds involving hydrogen atoms. A simulation time step of 2 fs was used along with the MD simulation. The Langevin thermostat [27] with a collision frequency of 2 ps^−1^ was employed for temperature control, while the Berendsen barostat [28] with a pressure-relaxation time of 1 ps was used to maintain the standard pressure of the system. The simulated models are then heated up to 310 K for 100 ps and are continuously held at this temperature for another 500 ns or until the simulations have reached equilibrium [43], which means the complexes were stable during the simulations. Each complex was simulated three independent MD runs by the difference velocity. Finally, the CPPTRAJ [44] was used to calculate the root-mean-square deviation (RMSD), number of contact atoms, intermolecular hydrogen bonding between ligand/3Cpro. In addition, the percentage of hydrogen bond occupation, binding pattern, and binding free energy of the most efficient ligand against the two enzymes was further analyzed and compared with rupintrivir.

## 4. Conclusions

This work provided the newly designed rupintrivir analog P2-m3 with enhanced binding efficiency. By the aminomethyl substitution, this compound showed more hydrogen bonds than rupintrivir with L127 and K130 residue at the P2 site of CV-A16 and EV-A71 3Cpro. A moderate hydrogen bonding with G164 (N^3^) at P3 was found in CV-A16 3Cpro. Relative to rupintrivir, there was a more significant contribution from the additional key residues for P2-m3 binding, i.e., F25, N126, S128, and K130 in EV-A71 and F25, S128, K130, K143, Q146, and H161 in CV-A16. Altogether, this leads to a better binding affinity of such novel rupintrivir derivative P2-m3 as predicted by the MM-PBSA method. The P2-m3 was suggested to be synthesized and tested for further development as the anti-HFMD agent.

## Figures and Tables

**Figure 1 molecules-27-01908-f001:**
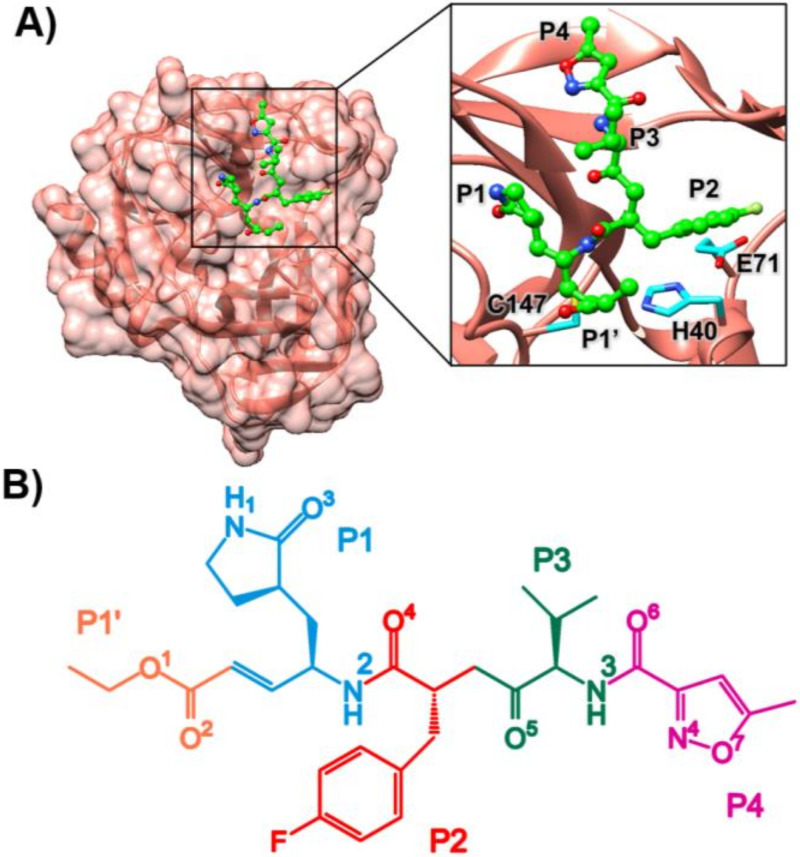
(**A**) The 3D structure of EV-A71 3Cpro in complex with rupintrivir (PDB ID: 3R0F) [27] (ball and stick green model), where the catalytic triad are shown in blue stick model. (**B**) The chemical structure of rupintrivir.

**Figure 2 molecules-27-01908-f002:**
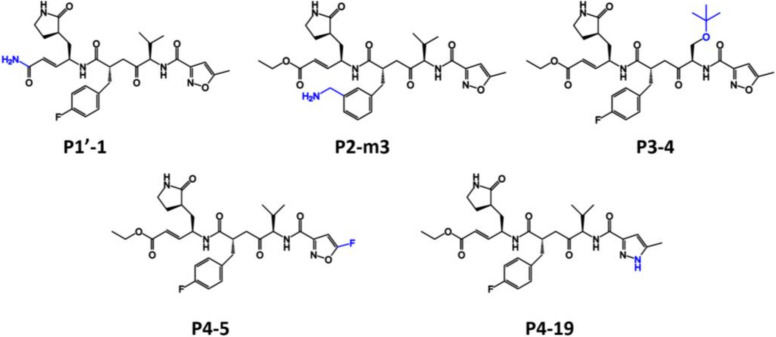
The chemical structure of five selected compounds with $\Delta \Delta G_{bind}$ < 0 kcal/mol.

**Figure 3 molecules-27-01908-f003:**
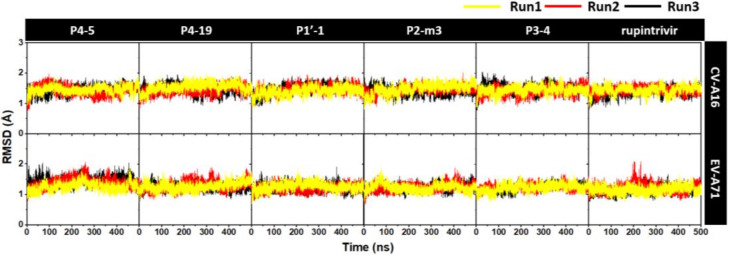
All-atom RMSD plots for the CV-A16 and EV-A71 3Cpro in complex with five focused analogs P4-5, P4-19, P1′-1, P2-m3, and P3-4, as well as rupintrivir, plotted along the 500 ns from the three independent simulations (Run1, Run2, and Run3).

**Figure 4 molecules-27-01908-f004:**
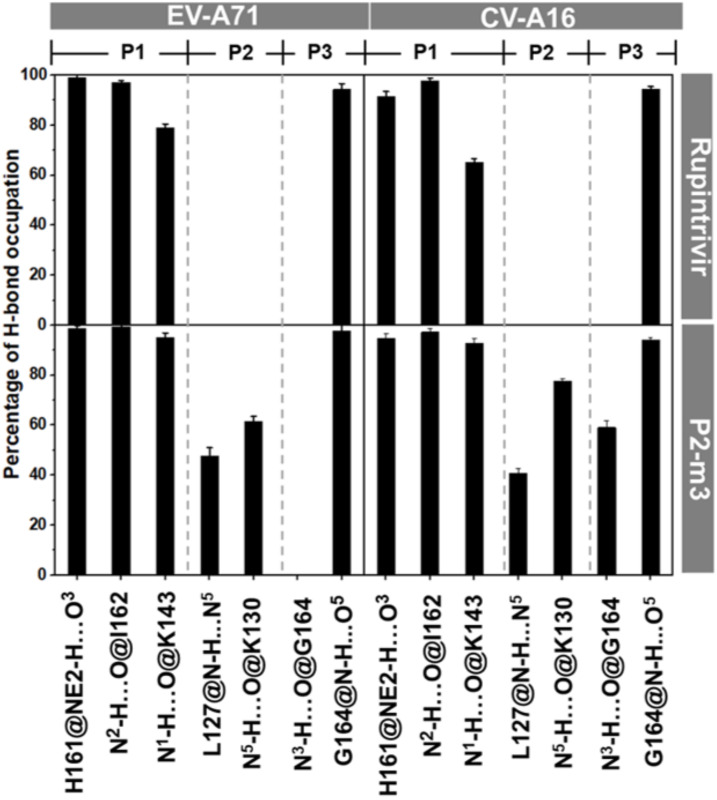
Percentage of intermolecular H-bond occupation with P1, P2, and P3 sites of the rupintrivir and its analog P2-m3 with EV-A71 and CV-A16 3Cpro derived from the last 50 ns simulations, where the representative structures are depicted in Figure 5. Only H-bond occupation > 40% is shown in the histogram.

**Figure 5 molecules-27-01908-f005:**
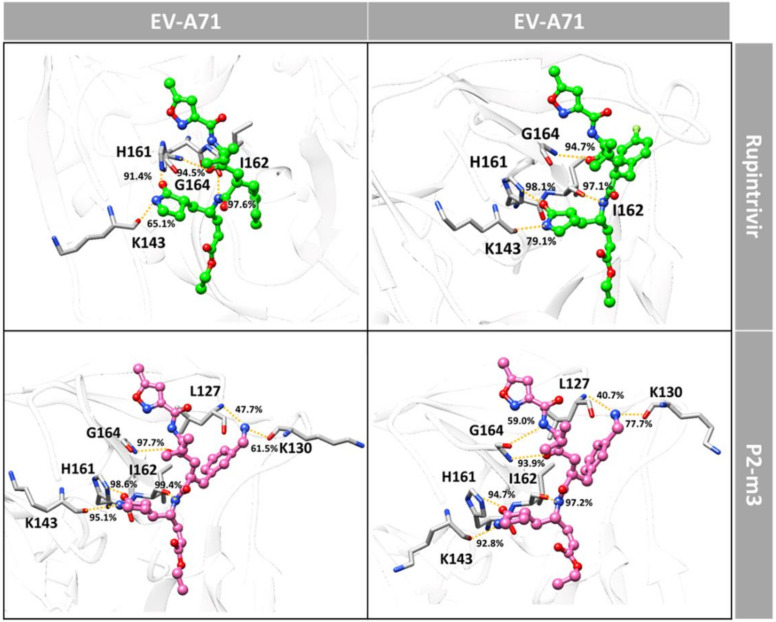
Hydrogen bonding interactions of the rupintrivir and its analog P2-m3 (bond and stick model) with EV-A71 and CV-A16 3Cpro residues (stick model).

**Figure 6 molecules-27-01908-f006:**
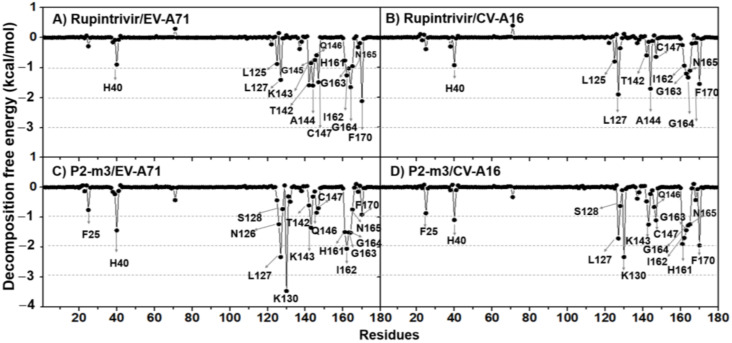
MM-PBSA per-residue decomposition free energy of the rupintrivir and its analog P2-m3 in complex with EV-A71 and CV-A16 3Cpro.

**Figure 7 molecules-27-01908-f007:**
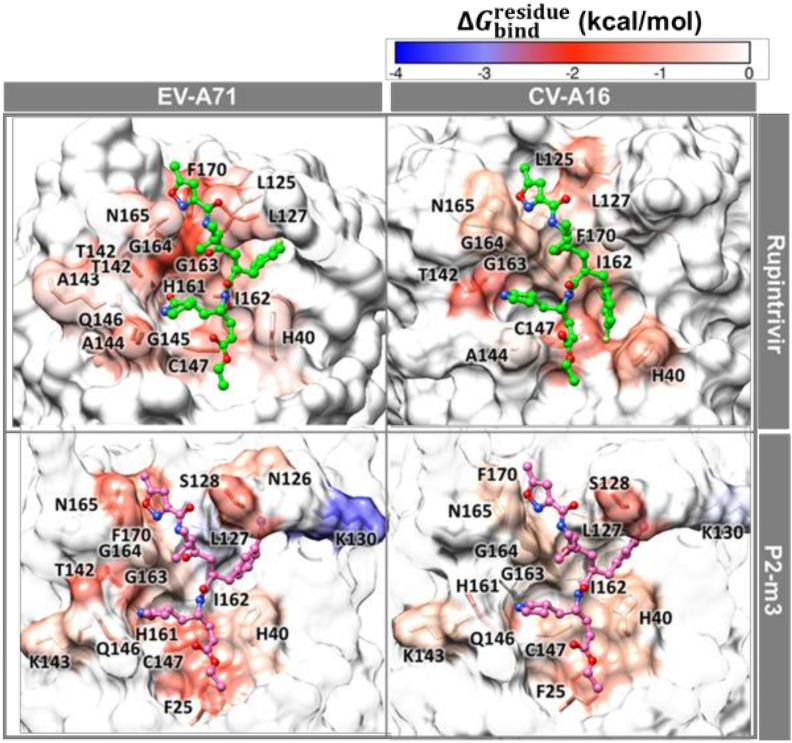
The contributing residues involved in ligand binding are colored according to the per-residue decomposition free energy ($\Delta G_{\mathrm{bind}}^{\mathrm{residue}}$), where the highest to lowest free energies are shaded from white to blue.

**Figure 8 molecules-27-01908-f008:**
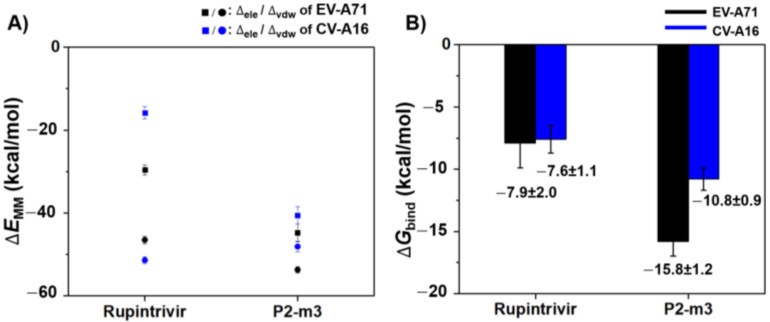
(**A**) The molecular mechanical energy ($\Delta E_{\mathrm{MM}}$) including electrostatic ($\Delta E_{\mathrm{ele}}$ ) and van der Waals ($\Delta E_{\mathrm{vdw}}$ ) interactions and (**B**) binding free energy ($\Delta G_{bind}$ ) based on the MM/PBSA method for rupintrivir and P2-m3 binding to EV-A71 and CV-A16 3Cpro.

**Table 1 molecules-27-01908-t001:** Relative interaction energy ($\Delta \Delta G_{bind}$) of the designed compounds in comparison to rupintrivir against EV-A71 3Cpro derived from MM/PBSA and MM/GBSA methods ($\Delta G_{bind}^{rupintrivir}$ of −19.38 and −32.83 kcal/mol, respectively). The compounds with negative $\Delta \Delta G_{bind}$ are in bold, and their 2D structures are given in Figure 2.

Compound	$\Delta \Delta G_{bind}(\mathrm{kcal}/\mathrm{mol})$		Compound	$\Delta \Delta G_{bind}(\mathrm{kcal}/\mathrm{mol})$	
	MM/PBSA	MM/GBSA		MM/PBSA	MM/GBSA
**P1′-1**	−9.14	−6.5	P2-m9	0.4	0.94
P1′-2	0.17	4.98	P2-m10	3.25	8.96
P1-1	7.82	5.02	P3-1	0.43	3.24
P1-2	20.41	17.58	P3-2	5.67	7.82
P1-3	1.37	1.91	P3-3	2.54	4.15
P1-4	8.28	4.21	**P3-4**	−4.9	−2.54
P1-5	17.76	14.31	P4-1	1.98	6.27
P2-p1	6.38	5.49	P4-2	1.73	4.25
P2-p2	14.17	14.15	P4-3	2.28	7.36
P2-p3	1.48	2.14	P4-4	12.74	12.79
P2-p4	7.55	9.22	**P4-5**	−5.34	−5.68
P2-p5	7.85	11.03	P4-6	4.46	2.51
P2-p6	0.73	3.55	P4-7	4.65	4.29
P2-p7	1.37	5.18	P4-8	3.32	2.94
P2-p8	6.65	7.74	P4-9	6.58	4.92
P2-p9	0.65	1.38	P4-10	0.75	1.27
P2-p10	1.56	3.93	P4-11	4.89	3.34
P2-m1	20.47	18.36	P4-12	13.28	11.89
P2-m2	18.59	17.29	P4-13	1.57	0.47
**P2-m3**	−12.12	−4.2	P4-14	1.64	0.96
P2-m4	2.41	6.29	P4-15	12.68	11.51
P2-m5	8.74	12.38	P4-16	1.55	1.12
P2-m6	1.75	1.35	P4-17	3.03	2.52
P2-m7	0.49	2.03	P4-18	4.12	4.44
P2-m8	1.98	4.71	**P4-19**	−5.55	−6.68

**Table 2 molecules-27-01908-t002:** Number of contact atoms within the 3.5-Å sphere of the focused compounds for CV-A16 and EV-A71 3Cpro systems taken from the last 50 ns of three independent simulations. The analog with bold was a potent compound.

	CV-A16	EV-A71
P4-5	22.5 ± 5.0	16.1 ± 4.7
P4-19	22.1 ± 5.3	17.4 ± 4.8
P1′-1	19.3 ± 4.3	17.8 ± 4.5
**P2-m3**	**24.3 ± 4.3**	**22.1 ± 4.5**
P3-4	17.9 ± 4.8	21.6 ± 4.8
Rupintrivir	23.7 ± 4.7	20.0 ± 5.0

**Table 3 molecules-27-01908-t003:** Number of hydrogen bonds of the focused compounds with CV-A16 and EV-A71 3Cpro calculated from the last 50 ns of the three independent simulations.

	CV-A16	EV-A71
P4-5	4.8 ± 1.1	3.6 ± 1.3
P4-19	4.0 ± 1.2	4.4 ± 1.2
P1′-1	4.0 ± 1.0	5.1 ± 1.1
**P2-m3**	**6.5 ± 1.2**	**5.6 ± 1.5**
P3-4	3.3 ± 1.3	4.8 ± 1.1
Rupintrivir	4.5 ± 0.9	4.7 ± 1.1

## Data Availability

Not applicable.

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
