# Peer review of "Computational Screening of Newly Designed Compounds against Coxsackievirus A16 and Enterovirus A71"

_molecules, 2022, doi:10.3390/molecules27061908_

Round 1
Reviewer 1 Report
Amita Sripattaraphan et.al aimed to find a me-better compound of Rupintrivir against CV-A16 and EV-A71 3CPro through virtual screening. P2-m3 was eventually find with greatest potentials to interact with the 3Cpro protein through delivering the highest number of intermolecular hydrogen bonds and contact atoms. Although the results of the virtual screening are perfect, they need to be biologically validated to make sense. In a word, the biggest shortcoming of this paper is the lack of biological experimental verification. Therefore, I suggest the author to supplement relevant experiments and optimize based on these results.
Author Response
Referee’s 1 comments:
Amita Sripattaraphan et.al aimed to find a me-better compound of Rupintrivir against CV-A16 and EV-A71 3CPro through virtual screening. P2-m3 was eventually find with greatest potentials to interact with the 3Cpro protein through delivering the highest number of intermolecular hydrogen bonds and contact atoms. Although the results of the virtual screening are perfect, they need to be biologically validated to make sense. In a word, the biggest shortcoming of this paper is the lack of biological experimental verification. Therefore, I suggest the author to supplement relevant experiments and optimize based on these results.
Response: Thank you for the reviewer’s suggestion. The P2-m3 analog showed the greatest potential to interact with CV-A16 and EV-A71 3Cpro from computational studies. Therefore, this compound needs to be synthesized before applying it to the antiviral assay. Since we have time and budget limitations to synthesize; therefore, we cannot synthesize the compound in this work. However, the biological experiment of synthesizing P2-m3 analog could be performed in further study to validate the biological efficiency against CV-A16 and EV-A71.
Reviewer 2 Report
Reviewer’s report
Manuscript:
Computational Screening of Newly Designed Compounds against Coxsackievirus A16 and Enterovirus A71
In this work, the binding free energy prediction and all-atom molecular dynamics (MD) simulation were employed to screen a series of the designed rupintrivir analogs against 3Cpro of CV-A16 and 82 EV-A71.
- The authors mentioned in the introduction that “rupintrivir failed to meet desired clinical parameters and also it was poorly aqueous soluble with low oral bioavailability challenging further pharmacological development”
Could the authors explain more the choice of Rupintrivir as an antiviral since it couldn’t achieve the last phases of trials?
- Introduction lines 39-40 please check the reference 1, which is related to rubella virus in China not HFMD
- Reference 9, 37, 39, 40 and 42 are not mentioned in the text
- line 63: Picornaviridae should be in italic Picornaviridae
Author Response
Referee’s 2 comments:
In this work, the binding free energy prediction and all-atom molecular dynamics (MD) simulation were employed to screen a series of the designed rupintrivir analogs against 3Cpro of CV-A16 and 82 EV-A71.
Response: First, we would like to thank referee for positive comments and suggestions.
Comment 1: The authors mentioned in the introduction that “rupintrivir failed to meet desired clinical parameters and also it was poorly aqueous soluble with low oral bioavailability challenging further pharmacological development” Could the authors explain more the choice of Rupintrivir as an antiviral since it couldn’t achieve the last phases of trials?
Response 1: The rupintrivir is a synthetic compound that is a member of the protease inhibitor class of antiviral agents. It is designed specifically for rhinovirus for the first time and later it become a candidate drug for the virus in the family of Picornaviridae including EV-A71 and CV-A16. Even though the rupintrivir showed a lack of efficacy in natural infection and was poorly aqueous soluble with low oral bioavailability, many studies reported that rupintrivir also shows a broad-spectrum antibiotic against other members of the family Picornaviridae that encodes 3C or 3C-like protease. More details have been mentioned in lines 59-70. Therefore, the solubility of the rupintrivir analogs was improved in this work. From the ADMET properties prediction, the P2-m3 showed better aqueous solubility than rupintrivir. We have added the solubility result of ADMET property in lines 218-219 and supplementary information Fig. S4.
Comment 2: Introduction lines 39-40 please check reference 1, which is related to rubella virus in China not HFMD
Response 2: Thank you for your careful review. The references have been corrected in lines 297-299.
Comment 3: Reference 9, 37, 40, 39, and 42 are not mentioned in the text
Response 3: Thank you for your careful review. The references have been corrected in lines 43, 240, 243, and 82 for references 9, 37, 39, 40 and 42, respectively.
Comment 4: line 63: Picornaviridae should be in italic Picornaviridae
Response 4: We have corrected it in line 63.
Reviewer 3 Report
In their manuscript entitled 'Computational Screening of Newly Designed
Compounds Against Coxsackievirus A16 and Enterovirus A71 ' the authors around Rungrotmongkol perform an in-silico search of novel rupintrivir analogs as potential drug candidates agains 3C protease from Coxsackievirus-A16 and Enterovirus-A71. The five best candidates were further investigated by means of molecular dynamics simulations yielding a single best candidate as potential drug against the two proteases.
The work should be published after, however, several issues are addressed.
The major ones are:
1, The authors must provide the generated force field parameter files as
additional supplement in separate files; in that way it is ensured that
interesed readers may inspect and maybe reuse their parameters. This is
necessary for documentation. It is good scientific practice to also include the
Gaussian-optimized structures (in pdb, mol2, or xyz format).
2, The method applied for screening is *not* MMGBSA, nor are the
reported energies actual *Free* energies, most probably: The MMGB/SA method is a statistical method to compute a mean value of a large amount of structures, as the author did for their lead compound. Furthermore,
it normally needs an entropic contribution term, which is calculated
separately. Since the authors did not describe that, I doubt that the entropic
contribution has been computed.
Thus, the term "free binding energy" seems not justified here, the reported
screening energies are merely "interaction energies" that used the MMGBSA scheme.
The authors are kindly requested to change their wording accordingly.
3, However, the authors should use their generated MD data for postprocessing all of their simulated systems with MM/GBSA, in order to obtain statistically more meaningful binding energies. Thus, please add a Table with the MM/(GB)PBSA energies for all the 2x6 systems, e.g. using the 50ns from each separate run.
The minor ones include:
- Fig 1.: Please give PDB id and reference for the depicted structure.
- Fig. 2A: Convert to Table, so the reader can compare the values more easily.
- Fig. 2B: Please indicate correct stereochemistry in the Lewis structures.
- Fig. 3: The different MD run curves are hard to discern due to the similar
color; please use actual colors (like black, red, yellow).
The protein RMSD values (Fig 3) are very low indicating a very stable
system, nice! However, it is necessary to also analyse whether the ligand keeps its binding pose; thus please add (to the supplement) a Figure similar to Fig. 3, where the protein systems are fit onto the first frame and the all-atom-RMSD for the ligand is computed without fitting.
- In the introduction (l. 70-77) it is unclear from the location of the
references, from which workgroup the suggestion came.
- Table S1: Orientation of rupintrivir is misleading; better use same orientation as in Table. Please add correct stereochemistry in the structures in the table.
- (Rational design):
Where do the structural variations come from, i.e. which sidechains the
authors used for their purpose? Was there a library used? Is it all just
chemical intuition?
- System preparation:
* It should be clearly stated that the authors used gaff (gaff2?) as force field
for the ligands; add a reference to the supplement here.
* References [31-33] are by no means the standard references for parameter
development in Amber! Authors are kindly requested to cite the correct
original literature references.
* Please add the original references for ff14SB, and MMPBSA (add
parameters used here like GB-model and SA-parameters).
* Line 240: "or until the simulations haved reached equilibrium."
Please explain and provide references.
* Please add the information here, that three independent MD runs were
performed (how were set up to differ?).
* The authors may want to explain, why they used EV-A71 as reference protein for their initial energy filtering.
- Gaussian reference: Please include full author list (maybe in supplement)
- Amber reference: Please include full author list
Author Response
Referee’s 3 comments:
The work should be published after; however, several issues are addressed.
Response: Thank-you for your valuable comments and suggestions that allow us to greatly improve the quality of our manuscript.
Comment 1: The authors must provide the generated force field parameter files as additional supplement in separate files; in that way it is ensured that interesed readers may inspect and maybe reuse their parameters. This is necessary for documentation. It is good scientific practice to also include the Gaussian-optimized structures (in pdb, mol2, or xyz format).
Response 1: The force field parameters of P2-m3 including frcmod and prepin files have been attached to the revision file.
Comment 2: The method applied for screening is *not* MMGBSA, nor are the reported energies actual *Free* energies, most probably: The MMGB/SA method is a statistical method to compute a mean value of a large amount of structures, as the author did for their lead compound. Furthermore, it normally needs an entropic contribution term, which is calculated separately. Since the authors did not describe that, I doubt that the entropic contribution has been computed. Thus, the term "free binding energy" seems not justified here, the reported screening energies are merely "interaction energies" that used the MMGBSA scheme. The authors are kindly requested to change their wording accordingly.
Response 2: Thank-you for this suggestion. The entropic contribution has been computed in MM/PB(GB)SA method. I agree with your suggestions; therefore, the word “free binding energy” has been changed to “interaction energy” (lines 81, 105, 106, 121, 247, 248, and 253).
Comment 3: However, the authors should use their generated MD data for post processing all of their simulated systems with MM/GBSA, in order to obtain statistically more meaningful binding energies. Thus, please add a Table with the MM/(GB)PBSA energies for all the 2x6 systems, e.g., using the 50ns from each separate run.
Response 3: The result of MM/(GB)PBSA energies for rupintrivir and P2-m3 against CV-A16 and EV-A71 3Cpro in each system have been added in the Supplementary Information (Table S2).
The minor ones include:
Comment 3.1: Fig 1 Please give PDB id and reference for the depicted structure.
Response 3.1: The PDB ID and reference have been added in Fig 1 (line 87).
Comment 3.2: Fig. 2A: Convert to Table, so the reader can compare the values more easily.
Response 3.2: The relative interaction energy has been converted to Table 1 (line 125).
Comment 3.3: Fig. 2B Please indicate correct stereochemistry in the Lewis structures.
Response 3.3: We have corrected it in line 126.
Comment 3.4: Fig. 3: The different MD run curves are hard to discern due to the similar color; please use actual colors (like black, red, yellow). The protein RMSD values (Fig 3) are very low indicating a very stable system, nice! However, it is necessary to also analyse whether the ligand keeps its binding pose; thus please add (to the supplement) a Figure similar to Fig. 3, where the protein systems are fit onto the first frame and the all-atom-RMSD for the ligand is computed without fitting.
Response 3.4: Fig 3 has been changed as your suggestions. In addition, the superimposition of compounds, P1’-1, P2-m3, P3-4, P4-5, P4-19, and rupintrivir against CV-A16 and EV-A71 at the binding site derived from the last 50 ns of simulation have been added in Fig S2. It was found that suggested that all ligands were likely stable along with the simulations in the active site. This result has been added to the revised manuscript (lines 133-137).
Comment 3.5: In the introduction (l.70-77) it is unclear from the location of the references, from which workgroup the suggestion came.
Response 3.5: Thank you for this useful suggestion. The reference and more detail has been added in line 72.
Comment 3.6: Table S1: Orientation of rupintrivir is misleading; better use same orientation as in Table. Please add correct stereochemistry in the structures in the table.
Response 3.6: We have corrected the structures in Table S1 as your suggested.
Comment 3.6: (Rational design) Where do the structural variations come from, i.e. which sidechains the authors used for their purpose? Was there a library used? Is it all just chemical intuition?
Response 3.6: The reasons for the compound design have been mentioned in lines 93-100 as follows “The 50 analogs were modified using a structure-based drug design as follows. P1 was modified at O3 (e.g., chlorine, fluorine, and methanol) to better interact with K143 in the S1 subsite. P2 was enlarged by the bulkier side chain (e.g., methanamine, ethylamine, and ethanol) to consequently shorten the distance to K130 and Q71 residues in the S2 subsite. To interact with S128 in the S3 subsite, the addition of 2-propanol, 2-fluoropropane, or ammonia was introduced on the side chain of P3. As L125 is located in the S4 subsite, we decided to increase the length of the P4 side chain or to change the functional group (e.g., hydroxyl, methyl, and fluorine).”
Comment 3.7: System preparation:
Comment 3.7.1: It should be clearly stated that the authors used gaff (gaff2?) as force field for the ligands; add a reference to the supplement here.
Response 3.7.1: The ligand force field has been corrected in line 240.
Comment 3.7.2: References [31-33] are by no means the standard references for parameter development in Amber! Authors are kindly requested to cite the correct original literature references.
Response 3.7.2: The original references have been added in lines 381-386.
Comment 3.7.3: Please add the original references for ff14SB, and MMPBSA (add parameters used here like GB-model and SA-parameters).
Response 3.7.3: The original reference for ff14SB [reference 34] has been added (lines 237 and 387). For MMPBSA, we used the general parameters implemented in the AMBER16 package.
Comment 3.7.4: Line 240: "or until the simulations have reached equilibrium." Please explain and provide references.
Response: The meaning of “or until the simulations have reached equilibrium” is that the complexes were stable during the simulations, which could evaluate from the RMSD result. The RMSD of the complexes did not fluctuate, which means the systems have reached equilibrium. Explanation and reference 43 have been added in lines 262-263 and 406-408.
Comment 3.7.5: Please add the information here, hat three independent MD runs were performed (how were set up to differ?).
Response3.7.5: The 5 screened compounds and rupintrivir with EV-A71 and CV-A16 3Cpro were simulated independently three times by the difference velocity (lines 263-264).
Comment 3.7.6: The authors may want to explain, why they used EV-A71 as reference protein for their initial energy filtering.
Response 3.7.6: From the data that was reported in many studies, they reported that EV-A71 was the cause of severe and fatal cases of HFMD (90%) while non-EV-A7 enteroviruses were associated with less than 10% of severe and fatal cases (Meng, et al. 2020). Therefore, the EV-A71 was used as a reference protein for their initial energy filtering. We have added this information into the results part (line 102-104) and reference part [44] (line 409-411)
Comment 3.8: Gaussian reference: Please include full author list
Response3.8: The Gaussian reference has been corrected in reference 34 (line 387).
Comment 3.9: Amber reference: Please include full author list
Response 3.9: The Amber reference has been corrected in reference 35 (line 388-389).
Round 2
Reviewer 1 Report
In my opinion, the authors have not addressed my concerns and it's not suitable for publication in Molecules.